# Novel Broccoli Sulforaphane-Based Analogues Inhibit the Progression of Pancreatic Cancer without Side Effects

**DOI:** 10.3390/biom10050769

**Published:** 2020-05-15

**Authors:** Christina Georgikou, Laura Buglioni, Maximilian Bremerich, Nico Roubicek, Libo Yin, Wolfgang Gross, Carsten Sticht, Carsten Bolm, Ingrid Herr

**Affiliations:** 1Section Surgical Research, Molecular OncoSurgery, Department of General, Visceral and Transplantation Surgery, University of Heidelberg, 69120 Heidelberg, Germany; christine.georgikou@outlook.com (C.G.); yinlibo001@163.com (L.Y.); WGross@uni-hd.de (W.G.); 2Institute of Organic Chemistry, RWTH Aachen University, 52056 Aachen, Germany; l.buglioni@tue.nl (L.B.); maximilian.bremerich@rwth-aachen.de (M.B.); nico.roubicek@rwth-aachen.de (N.R.); 3Medical Research Center, Medical Faculty Mannheim, University of Heidelberg, 69117 Mannheim, Germany; carsten.sticht@medma.uni-heidelberg.de

**Keywords:** bioactive agents, drug development, microRNA signaling, pancreatic cancer, NCI-60, sulforaphane, sulfoximine

## Abstract

The naturally occurring isothiocyanate sulforaphane, found in *Brassicaceae* vegetables, is promising in cancer treatment, e.g., by the normalization of enhanced levels of NF-κB-signaling in tumor stem cells. We chemically synthesized seven sulforaphane analogues by substitution of the sulfinyl group (S(O)) to either sulfimidoyl (S(NR)) or sulfonimidoyl (S (O) (NR)) groups, and characterized them in the cell lines of pancreatic cancer and several other tumor entities, including the NCI-60 cell panel. MTT and colony forming assays, flow cytometry, immunohistochemistry, microRNA arrays, bioinformatics, tumor xenotransplantation, and Kaplan Meier survival curves were performed. Compared to sulforaphane, the analogue **SF102** was most efficient in inhibition of viability, colony formation, tumor growth, and the induction of apoptosis, followed by **SF134.** Side effects were not observed, as concluded from the body weight and liver histology of chick embryos and survival of *C. elegans* nematodes. Among 6659 differentially regulated microRNAs, miR29b-1-5p, and miR-27b-5p were downregulated by sulforaphane compared to controls, but upregulated by **SF102** and **SF134** compared to sulforaphane, suggesting differential signaling. Each substance was involved in the regulation of several NF-κB-related target genes. In conclusion, sulforaphane analogues are promising for the development of highly active new drugs in cancer treatment.

## 1. Introduction

Pancreatic ductal adenocarcinoma is the fourth leading cause of cancer-related death in the US, with a 5-year relative survival of 9% calculated for all stages, and of 3% when the tumor has spread [1,2]. The development of new therapeutic options is urgently needed.

Epidemiological studies indicate that higher consumption of *Brassicaceae* vegetables is correlated with a lower cancer risk [3]. Recent pilot studies with sulforaphane-enriched broccoli sprout supplementation in patients with advanced pancreatic or prostate cancer showed promising results [4,5]. *Brassicaceae* are unique in their high content of glucosinolates [6]. A focus has been placed on the glucosinolate glucoraphanin, which is found in high concentration in broccoli and its sprouts. Glucoraphanin is converted to its active form, the isothiocyanate sulforaphane [1-isothiocyanato-(4*R*)-(methylsulfinyl)-butane], by myrosinase-mediated hydrolysis [7]. Sulforaphane is one of the best studied bioactive anti-inflammatory agents and suggested to induce detoxifying enzymes, cell cycle arrest, apoptosis and epigenetic regulation [6]. In pancreatic cancer, sulforaphane normalized increased NF-κB signaling in tumor stem cells by the induction of miRNA-365 expression, which inhibited the expression of the NF-κB subunit c-Rel by binding to its 3´-UTR [8,9].

The development of highly active sulforaphane analogues suited for medication is urgently required. However, the establishment of sulforaphane analogues by molecular single atom changes in form of oxygen-to-nitrogen substitutions at the central sulfur, are unprecedented in the context of sulforaphane chemistry. This is surprising, considering that a significant number of other structural variations of sulforaphane are known [10,11,12,13], and that such one-atom modifications have recently led to significant improvements in other medically relevant areas [14,15,16,17,18,19,20].

In the present study, we prepared seven analogues of sulforaphane by formal substitutions of the sulfinyl group by sulfimidoyl or sulfoximidoyl moieties. The R substituent at the nitrogen was acetyl, pentafluorobenzoyl, methyl, trifluoroacetyl, or carbamoyl. All products represented monoaza analogues of sulforaphane or its sulfone, and were prepared as racemates. The analogue **SF102** was most effective in inhibition of viability, clonogenicity, and tumor growth, along with induction of apoptosis in cell lines of various tumor entities. No obvious side effects were observed. miRNA array profiling revealed differential signaling of sulforaphane, **SF102**, and **SF134**, but the regulation of NF-κB-related target genes was common.

## 2. Materials and Methods

### 2.1. Chemical Syntheses of Sulforaphane Analogues

We prepared 7 monoaza racemate analogues of sulforaphane (**SF85** and **SF101**) or its sulfone (**SF86**, **SF102**, **SF113**, **SF134**, and **SF135**), which differ from the parent compound by formal substitutions of the sulfinyl (S = O) group by either sulfimidoyl (S(NR)) or sulfoximidoyl (S(O)(NR)) moieties. The R substituent at the nitrogen is acetyl (as in **SF85** with S(NAc) and **SF86** with S(O)(NAc)), pentafluorobenzoyl (as in **SF101** with S(NC(O)C_6_F_5_) and **SF102** with S(O)(NC(O)C_6_F_5_)), methyl (as in **SF113** with S(O)(NMe)), trifluoroacetyl (as in **SF134** with S(O)(NCOCF_3_)), and carbamoyl (as in **SF135** with S(O)(NCONH_2_)). The product purities were generally >95% (as determined by NMR spectroscopy or HPLC). For the two most active compounds, the following purities were found: **SF134** ≥ 99.5% (as single batch); **SF102** 95.6%, 96.0%, and 92.3% (as three batches). The detailed description is provided (Appendix A).

### 2.2. Human Tumor Cell Lines

Established cell lines from pancreas (BxPc-3, AsPC-1), ovary (OVM), prostate (PC3), breast (BT-20), colorectum (SW707), glioblastoma (A172), liver (HepG2), neuroblastoma (IMR5), and T cell leukemia (Jurkat) were obtained from ATCC (Manassas, VA, USA). Cervical (P5) and lung (P693) cell lines and gemcitabine-resistant pancreatic cancer BxGEM cells are described [21,22]. Jurkat cells were cultured in RPMI-1640 medium, supplemented with 10% heat-inactivated FCS (both from Sigma-Aldrich, Deisenhoffen, Germany) and 25 mmol/L HEPES (PAA, Pasching, Austria). All other cancer cell lines were cultured in a DMEM/high glucose medium, supplemented with 10% heat-inactivated FCS (Sigma-Aldrich) and 25 mmol/L HEPES (PAA). Mycoplasma detection tests were performed monthly by the use of PlasmoTest™ (InvivoGen, San Diego, CA, USA). 

### 2.3. Reagents

Lyophilized powder of D,L-Sulforaphane (Sigma Aldrich) and its analogues were dissolved in DMSO to stock concentrations of 100 mM. Aliquots were stored at –80 °C. Each stock was used only once after thawing. DMSO alone was used as vehicle control diluted 1:1000 or higher.

### 2.4. Cell Viability Assay

Cell viability was measured using 3-(4,5-dimethylthiazol-2-yl)-2,5-diphenyltetrazolium bromide (MTT), as previously described [23].

### 2.5. Colony Forming Assay

The cells were seeded at a density of 2–4 × 10^4^ cells/well in 6-well cell culture plates. The cells were treated, trypsinized 24 h later and analyzed as described [23].

### 2.6. NCI-60 Human Tumor Cell Lines Screen

This screen was performed by the NCI, Division of Cancer Treatment & Diagnosis (https://dtp.cancer.gov/discovery_development/nci-60/default.htm). It utilized 60 human tumor cell lines to characterize the growth inhibitory or killing activities of a therapeutic subtstance. 

### 2.7. Tumor Xenotransplantation on Fertilized Chicken Eggs

The tumor xenotransplantation to genetically identical hybrid Lohman Brown chicken eggs was performed as previously described [24]. On day 9 of embryonic development, 7 × 10^5^ BxPc3 or BxGEM cells were transplanted at a ratio of 1:1 with Matrigel onto the chorioallantoic membrane (CAM) of each egg. Five days later, sulforaphane, its analogues were injected into CAM vessels (50 µL/egg, diluted in PBS). Control eggs were injected with 50 µL PBS alone. The xenografts were resected on day 18, and the tumor volumes were evaluated 3-dimensionally by a USB microscope camera (eScope, Oitez, Hongkong) and digital image editing, using a customized mount. 

### 2.8. Immunohistochemical Staining

Staining was performed on 6-µm frozen tissue sections from the tumor xenografts, as previously described [24]. A mouse monoclonal antibody against Ki67 (ab8191, 1:400, Abcam, Cambridge, UK) was used, and positive signals were quantified with the ImageJ software (NIH, Bethesda, MD, USA). 

### 2.9. Caenorhabditis Elegans Survival

*Caenorhabditis elegans* (*C. elegans*) N2 nematodes (CGC, Minnesota, USA) were kept at 20 °C. L4-stage worms were picked (n = 100/group) and transferred into 60 mm medium-sized plates (TPP, Trasadingen, Switzerland) containing S Basal medium, supplemented with *E. coli* OP50 bacteria [25]. Sulforaphane and analogues (400 µM), or DMSO alone, were added to S Basal medium for 48 h. Live worms were transferred daily to new plates until they stopped laying eggs. Worms that died of causes other than aging, such as internal hatching or vulva protrusion, were excluded. The survival rates were documented by Kaplan-Meier curves. 

### 2.10. miRNA Microarray Profiling

The miRNeasy Mini Kit was used for miRNA isolation, according to the manufacturer’s instructions (Qiagen, Hilden, Germany). The microarray analysis was performed at the Microarray-Analytic Center (Medical Faculty, Mannheim, Germany), using the Affymetrix GeneChip miRNA 4.0 Array with a total of 30,424 microRNAs (Thermo Fisher Scientific, Dreieich, Germany).

### 2.11. miRNA In Silico Analysis

The 500 most significantly differentially regulated miRNAs (*p* < 0.05) were selected from the microarray raw data. Volcano plots and Venn diagrams were created by comparison of miRNA expression between sulforaphane and control, **SF102**, or **SF134** and between **SF102** and **SF134**; miRNAs with –log10 *p*-Values >2 were included. To predict the target genes of the 3 most promising miRNA candidates we used the online platform mirWalk (http://zmf.umm.uni-heidelberg.de/apps/zmf/mirwalk2/) and its Target Mining option. 

### 2.12. Statistical Analysis

The quantitative data are presented as the mean values with standard deviations from at least three independent experiments performed in triplicate. The significance was analyzed using Student’s t-test for parametric data corrected for multiple comparisons with the Bonferroni-Holm method. For evaluation of the Kaplan Meyer analysis, the standard chi-square-based log-rank test and for miRNA microarray, the JMP software (SAS, Heidelberg, Germany) were used. *p* < 0.05 was considered statistically significant.

## 3. Results

### 3.1. Chemical Synthesis of Sulforaphane Derivatives

The sulforaphane analogues **SF85**, **SF86**, **SF101**, and **SF102** were prepared by light-induced ruthenium-catalyzed imidations of erucin (1) and sulforaphane (2) (Figure 1A). Dioxalones 3a and 3b served as nitrene precursors [26,27,28]. In general, the imidation reactions involving 1 proceeded better than those with 2, providing sulfilimines **SF85** and **SF101** in higher yields (96% and 92%) than the sulfoximines **SF86** and **SF102** (62% and 76%).

The syntheses of the sulforaphane analogues **SF113**, **SF135**, and **SF134** started from *N*-Boc protected sulfide 4 and sulfoxide 8 (Figure 1B) [29]. A one-pot imidation/oxidation of sulfide 4 provided *N*-methyl sulfoximine 5 in 30% yield [30]. As byproduct, 50% of the corresponding sulfone was observed (not shown). Acidic Boc-cleavage of 5 was followed by Boa’s protocol for converting amines into isothiocyanates [31]. Employing carbon disulfide and di-*tert*-butyl dicarbonate, **SF113** was synthesized in 75% yield. **SF135** was prepared by a metal-free imidation of sulfide 4 to give *N*-cyano sulfilimine 6 in 71% yield [32]. Oxidation of 6 with *m*CPBA provided 59% of the corresponding sulfoximine 7. For 7, Boc-cleavage under acidic conditions was accompanied by hydrolysis of the *N*-cyano group resulting in an intermediate with an *N*-carbamoyl moiety (not shown) [33,34]. Following Boa’s protocol [31] as before led to sulforaphane analog **SF135** in 83% yield. Key step in the synthesis of **SF134** was a rhodium-catalyzed imidation of sulfoxide 8, which provided *N*-trifluoroacetyl-protected sulfoximine 9 in 65% yield [35]. Applying protocols in analogy to the ones detailed above for the conversion of a Boc-protected amine into an isothiocyanate [31] afforded **SF134** from 9% to 50% yield.

A streamlined and protecting group-free synthesis of the two most active derivatives (**SF102** and **SF134**) started from tetrahydrothiophene (10), which was alkylated with methyliodide (Figure 1C). The corresponding *S*-methylthiophenium salt was isolated as its tetrafluoroborate (11), and the subsequent ring-opening substitution with sodium azide afforded the (4-azidobutyl)(methyl)sulfide (12) [36]. One-pot oxoimination with iodosobenzene diacetate (PIDA)/ammonium carbamate furnished the NH-sulfoximine 13 [37], which was acylated to both the trifluoroacetyl- and perfluorobenzoyl-derivatives 14 and 15. The azido terminus of the respective precursor was then subjected to an in situ reduction (*n*Bu_3_P in case of 14, Zn/NH_4_Cl in case of 15), and the resulting amine was finally converted to targeted sulforaphane analogues **SF134** and **SF102**.

### 3.2. **SF102** and **SF134** Are Cytotoxic in All the Several Evaluated Tumor Cells

The pancreatic cancer cell lines BxPc3, BxGEM, and AsPC-1 were treated with sulforaphane or the analogues **SF85, SF86, SF101, SF102, SF113, SF134**, and **SF135**, in concentrations of 10 µM and 30 µM. Twenty-four hours after treatment, the cell viability was measured by MTT assay and documented by microscopy. Sulforaphane strongly reduced the viability in all cell lines, compared to solvent DMSO controls or untreated cells, as expected (Figure 2A, Appendix A). The effects of **SF102** were comparable to sulforaphane, whereas **SF134** was less potent. All other derivatives had no significant effects, and consequently, they were excluded from further testing.

To study the ability to interfere with clonogenicity, which is a typical tumor stem cell feature, BxPc-3 and AsPC-1 cells were treated with sulforaphane, **SF102** or **SF134**. After 24 h, a colony-forming assay was performed, and the number of surviving cells was evaluated by microscopy (Figure 2B). **SF102** and **SF134** significantly reduced the number of colonies even in chemoresistant AsPC-1 cells, although the effect of sulforaphane was more pronounced. In BxPc-3 cells, **SF102** was most potent in decreasing colony formation, followed by sulforaphane and **SF134**. To test if the effect on clonogenicity is long lasting, we isolated single live cells from the colonies, and re-seeded them without additional treatment. In the resulting second generation, the number of colonies was induced even stronger, suggesting that the treatment indeed eliminated the more aggressive, colony-forming, tumor stem cell-like cancer cells.

The observed therapeutic effects also occurred in cancer cells from cervix ovary, prostate, breast, colorectum, and lung, as well as in hepatocellular, neuroblastoma, T-cell leukemia, and glioblastoma cell lines, as detected by MTT assay 24 h after treatment (Figure 3). By the use of the NCI-60 cancer cell line panel, we confirmed the high potency of **SF102** in induction of growth inhibition and lethality in cancer cell lines of leukemia, melanoma, non-small-cell lung carcinoma, and cancers of the brain, ovary, breast, colon, kidney, and prostate (Figure 4). Likewise, **SF101** was effective in the NCI-60 panel test (Appendix A), which is consistent with the observed reduction of viability in pancreatic cancer cells (compare Figure 2A), although these data were not statistically significant. These results were confirmed in pancreatic cancer cells, where **SF102** most potentially induced apoptosis, even in resistant and highly aggressive BxGEM and AsPC-1 cells (Appendix A). 

### 3.3. **SF102** and **SF134** Inhibit Tumor Growth without Obvious Side Effects

To evaluate the in vivo relevance, we performed xenograft studies by transplanting BxPc-3 cells onto the CAM of fertilized chicken eggs on day 9 of chick development. Five days later, sulforaphane, **SF102**, **SF134**, or a PBS control were injected into CAM vessels, followed by the resection of the tumor xenografts on day 18. Although visual inspection showed an apparent tumor suppressing effect for sulforaphane, **SF102** and **SF134**, these differences were not significant (Figure 5A). To examine the proliferation, xenograft cryosections were stained with the marker Ki67, followed by immunohistochemistry and evaluation of the fluorescence intensity. Representative pictures and quantitative analysis demonstrated that **SF102** significantly reduced the proliferation to about 25% (Figure 5B), followed by sulforaphane and **SF134**, although the latter were not statistically significant.

To further investigate toxic side effects, the liver morphology of the chicken embryos and their body weight were analyzed at day 18 of development. As indicated by hematoxylin staining, no necrotic areas were detected in liver tissue (Figure 5C), and the body weight between 12 to 14 g was consistent between groups (Figure 5D). To further rule out any side effects, we used the *C. elegans* nematode model and treated the worms with sulforaphane, **SF102**, **SF134**, or the vehicle control DMSO for 48 h. The survival of the worms over a period of 35 days was evaluated by Kaplan Meier analysis. We did not observe a shorter survival of worms after treatment with sulforaphane, **SF102**, and **SF134** compared to the control (Figure 5E), suggesting that no apparent side effects regarding reproduction and development occurred.

### 3.4. Sulforaphane, **SF102** and **SF134** Induce Differential microRNA Expression and NF-κB Signaling

To further highlight the underlying signaling pathways responsible for the observed effects, we treated AsPC-1 cells with sulforaphane, **SF102**, or **SF134** or let them untreated. The RNA was isolated 24 h later and examined by miRNA microarray expression profiling, followed by bioinformatic evaluation. We identified 500 significantly (*p* < 0.05) differentially regulated miRNAs compared to the control. As presented in a heat map, there were many similar clusters but also obvious differences in miRNA expression following sulforaphane, **SF102**, and **SF134** treatment (Figure 6A). Next, the top 10 most significantly differentially regulated miRNAs were compared between control and sulforaphane, control and **SF102**, and control and **SF134** (Figure 6B). By this method, we identified miR2278 as common and most significantly downregulated miRNA following sulforaphane, **SF102**, and **SF134** treatment (Appendix A). To further highlight differences in miRNA expression, we created volcano plots from a comparison of the most significantly differentially regulated miRNAs with –log10 P values >2. The topmost interesting candidates were miR27b-5p and miR29b-1-5p, which were downregulated after sulforaphane treatment compared to the control, but upregulated by **SF102** and **SF134** compared to the sulforaphane (Figure 6C, Appendix A). The number of target miRNAs regulated by sulforaphane, **SF102**, and **SF134** is schematically presented in a Venn diagram, and the names of the identified miRNA candidates are provided in a table (Figure 6D, Appendix A).

To obtain information about target genes regulated by the three candidate miRNAs, a bioinformatics analysis using the mirWalk online platform and its Target Mining tool was performed. Based on the highest predicted binding probability of miRNAs to target gene sites, we identified 309 target genes that are individually or commonly regulated by the three miRNA candidates, as visualized in a network plot (Figure 7A). To follow up on our previous studies showing that sulforaphane normalizes enhanced NF-κB activity by miRNA signaling [8,23], several candidate genes were identified (Figure 7B, Appendix A). The top candidate was miR2278, because it was predicted to regulate the highest number of NF-κB-related target genes, followed by miR27b-5p and miR29b-1-5p, which partially overlapped in regulation of NF-κB-related target genes.

## 4. Discussion

Here, we prepared a series of unprecedented analogues of sulforaphane with sulfur core variations. We examined the anti-cancer potential of seven analogues and compared them to sulforaphane. Viability assays in pancreatic cancer cell lines allowed to identify the two most promising derivatives, **SF102** and **SF134**. They inhibited colony-formation, xenograft growth, and induced apoptosis in a variety of cancer cells of various tumor entities. **SF102** was partially more active than sulforaphane. Most importantly, obvious side effects in vivo were not observed. As detected by miRNA array analysis and bioinformatics evaluation, there were significant differences in signal transduction between sulforaphane, **SF102,** and **SF134**. However, all of them had in common the regulation of NF-κB-related target genes.

To the best of our knowledge, this is the first time that the cytotoxicity of synthetic sulforaphane analogues has been tested and evaluated in pancreatic cancer. Previously, other sulforaphane derivatives with modifications in the carbon substituents at the sulfur core were shown to inhibit the proliferation of melanoma cells [13], or to induce cell cycle arrest and apoptosis in hepatocellular carcinoma cells [38]. In this regard, Shi et al. demonstrated that sulforaphane heterocyclic analogues enhanced apoptosis and eliminated the tumor stem cell population in breast cancer cells [39]. These data are in line with our results, as the most effective **SF102** analogue did not only inhibit the viability of pancreatic cancer cells but also of breast, hepatocellular, and several cancer types, suggesting a broad range of anti-tumor activity.

Our results suggest that **SF102** and **SF134** decrease the viability, clonogenicity, and tumor growth, and induce apoptosis in pancreatic cancer cells as potent or, in the case of **SF134**, even better than sulforaphane. As already known, sulforaphane normalizes enhanced NF-κB signaling in tumor stem cells and thereby mediates chemosensitization [23], among several other activities. Our bioinformatic analysis of miRNA array data now predict that sulforaphane, **SF102**, and **SF134** affect NF-κB signaling by the induction of miR2278, which is involved in the regulation of many NF-κB target genes. Additionally, while sulforaphane inhibits the expression of miR27b-5p and miR29b-1-5p compared to untreated control cells, **SF102** and **SF134** rather induce these miRNAs compared to their expression in sulforaphane-treated cells. This finding is of high relevance, because, according to our in silico analysis, miR27b-5p and miR29b-1-5p are also regulators NF-κB-related target genes. According to the Human Protein Atlas, the predicted target genes PLCG1 (https://www.proteinatlas.org/ENSG00000124181-PLCG1/pathology) and TRIM25 (https://www.proteinatlas.org/ENSG00000121060-TRIM25/pathology) are of high relevance in pancreatic cancer, because patients with high PLCG1 (n = 110) or TRIM25 (n = 121) expression survive longer than those with low expression (n = 66, N = 55, respectively). Interestingly, our miRNA candidates miR2278, miR27b-5p and miR29b-1-5p have never before been documented in pancreatic cancer, as far as we know. However, Kim et al. demonstrated both miR27b-5p and miR29b-1-5p as markers for gastric cancer progression [40], while miR29b-1-5p overexpression induced epithelial-mesenchymal transition in oral squamous cell carcinoma [41]. The only available information about miR2278 and cancer is that its upregulation is associated with the inhibition of leukemic cell proliferation and enhanced apoptosis [42].

To confirm an in vivo activity of sulforaphane, we used tumor xenotransplantation to fertilized chicken eggs, an animal model that offers an ethical alternative to mammalian assays and has frequently been used in cancer research for xenotransplantation and drug delivery studies [43]. Our results suggest that the derivatives **SF102** and **SF134** decrease the tumor volume as sulforaphane, whereas **SF102** significantly inhibited the proliferation. Sulforaphane, **SF102**, and **SF134** caused no obvious side effects, as liver necrosis or abnormal weight loss of chicken embryos was not detectable. Our results obtained by the use of the animal model *C. elegans* demonstrated that sulforaphane and its analogues did not reduce the lifespan of the nematodes. In contrast, **SF102** even significantly induced longevity, a process that has been proven to be anticarcinogenic [44]. These observations could lead to the conclusion that **SF102** has the highest activity.

Regarding the clinical relevance of sulforaphane and its derivatives, a pilot study with freeze-dried, pulverized broccoli sprouts was performed in our Department of Surgery of the University Clinic Heidelberg. Forty patients were involved, suffering from advanced, non-resectable pancreatic cancer. The results indicated that the patients in the broccoli sprout group survived longer compared to the placebo group. However, these data were not significant, which might be due to the small number of patients [4]. Noteworthy, the daily intake of 15 capsules (90 mg/508 μmol) in order to reach a high sulforaphane concentration was unpleasant for the patients, because it increased pre-existing digestion problems in some patients and caused a higher drop-out rate. The development of a drug with highly active sulforaphane or a derivative thereof is urgently required to circumvent the unpleasant side effects of broccoli consumption, and to enable the intake of a high sulforaphane amount by a single capsule.

In conclusion, we identified two sulforaphane analogues, which showed promising inhibitory properties in all of several examined tumor entities. Although our data are convincing, we suggest taking them with caution as well. For example, product purity and stability must be regarded as critical factor, in particular, as we found in a previous study that different sulforaphane lots from a commercial supplier led to data deviations (Herr et al., unpublished data). Furthermore, trace impurities and byproducts stemming from the syntheses and compound degradations during their application might play a role. Notably, all compounds applied here—including sulforaphane, which in nature occurs as *(R)*-enantiomer—were racemic, and it is reasonable to assume that the use of single enantiomers would affect the inhibitory properties of the molecules. If so, the results would indeed vary. At the same time, however, new opportunities could arise by exclusively using the more effective stereoisomer allowing a dose reduction in subsequent treatments. Finally, a detailed analysis of the absorption, distribution, metabolism, excretion and toxicity (ADMET) properties of the new sulforaphane analogs is still lacking. Certainly, factors such as solubility and transport will significantly impact the biological activity of the compounds. Overall, we are convinced that chemical modifications of the sulforaphane scaffold are promising in opening unidentified opportunities for the future development of new sulforaphane-related drugs.

## Figures and Tables

**Figure 1 biomolecules-10-00769-f001:**
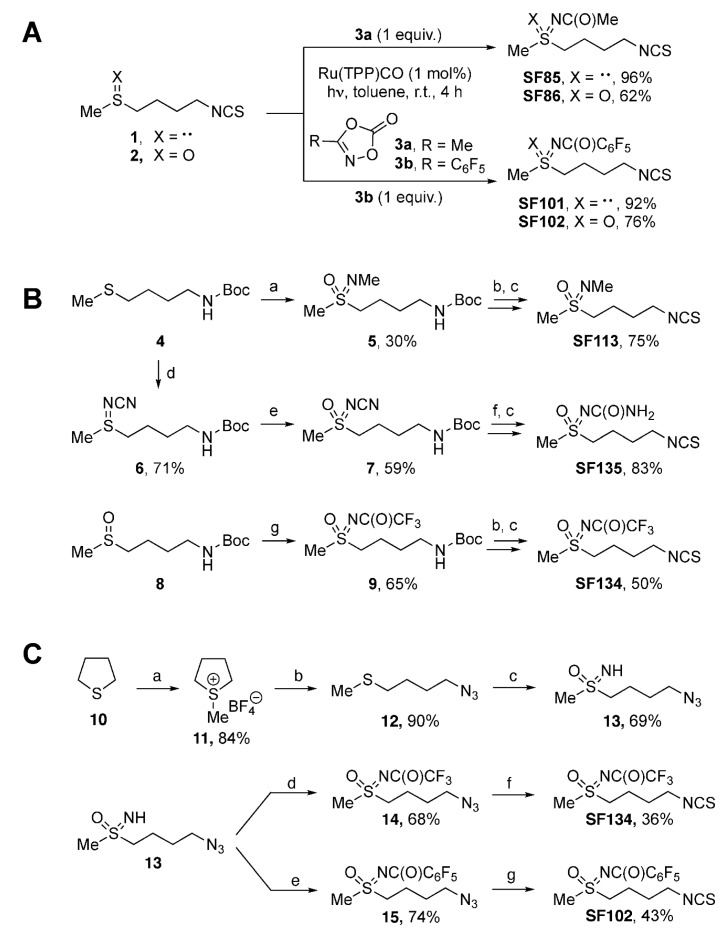
**Scheme of chemical synthesis of sulforaphane derivatives.** (**A**) Imidations of erucin (1) and sulforaphane (2) by light-induced ruthenium catalysis leading to the respective sulforaphane analogues **SF85**, **SF86**, **SF101**, and **SF102**. (**B**) Syntheses of sulforaphane analogues **SF113**, **SF135**, and **SF134**. Procedures: (a) 1. MeNH_2_ (3.0 equiv.); Br_2_ (2.0 equiv.), MeOH, r.t. 20 min.; 2. KMnO_4_ (3.0 equiv.), K_2_CO_3_ (2.0 equiv.), acetone, r.t., 12 h; (b) 4N HCl in dioxane (3.0 equiv.), dry DCM, r.t., 4 h; (c) 1. TEA (2.0 equiv.), CS_2_ (10.0 equiv.), dry EtOH, r.t., 60 min; 2. Boc_2_O (1.0 equiv.), DMAP (2 mol%), r.t., 60 min; NH_2_CN (1.5 equiv.), (d) PhI(OAc)_2_ (1.1 equiv.), CH_3_CN, r.t., 16 h; (e) *m*CPBA (1.5 equiv.), K_2_CO_3_ (3.0 equiv.), DCM; r.t., 16 h; (f) TFA (13.0 equiv.), Et_3_SiH (2.5 equiv.), DCM, r.t., 16 h; (g) Rh_2_(OAc)_4_ (2.5 mol%), CF_3_C(O)NH_2_ (2.0 equiv.), MgO (4.0 equiv.); PhI(OAc)_2_ (1.5 equiv.), DCM, r.t., 16 h. (**C**) Protecting group-free syntheses of the sulforaphane analogues **SF102** and **SF134**. Procedures: (a) 1. MeI (1.0 equiv.), 50 °C, 1 h, ultrasound, NaBF_4_ (1.0 equiv.), EtOH, reflux, 30 min; (b) NaN_3_ (3.0 eq.), DMF, 80 °C, 16 h; (c) PhI(OAc)_2_ (2.5 equiv.), ammonium carbamate (4.0 equiv.), MeOH, r.t., 90 min; (d) trifluoroacetic anhydride (1.2 equiv.), Et_3_N (1.3 equiv.), 4-DMAP (0.1 equiv.), DCM, r.t., 16 h; (e) perfluorobenzoyl chloride (1.5 equiv.), Et_3_N (1.5 equiv.), DCM, r.t., 16 h; (f) 1. *n*Bu_3_P (1.1 equiv.), THF, r.t., 30 min; 2. CS_2_ (3.0 equiv.), Et_3_N (3.5 equiv.), r.t., 90 min; 3. MsCl (1.2 equiv.), r.t., 40 min; (g) 1. Zn powder (10 equiv.), NH_4_Cl (2.5 equiv.), MeOH, r.t., 30 min; 2. CS_2_ (1.5 equiv.), Et_3_N (4.5 equiv.), THF, r.t., 90 min; 3. MsCl (1.1 equiv.), r.t., 40 min.

**Figure 2 biomolecules-10-00769-f002:**
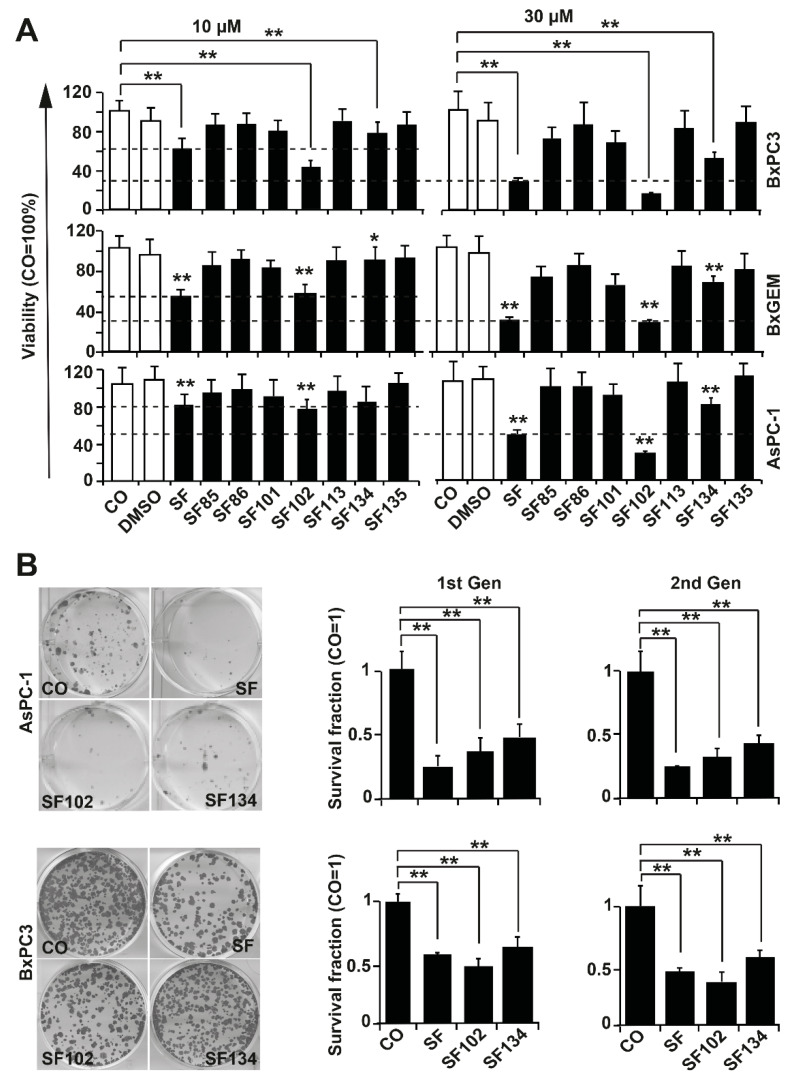
**SF102 and SF134 repress viability and colony formation.** (**A**) BxPc-3, BxGEM, and AsPC-1 cells were treated with sulforaphane (**SF**, 10 µM or 30 µM) and its analogues **SF85, SF86, SF101, SF102, SF113, SF134**, and **SF135**, or were left untreated (CO). After 24 h, the viability was measured by 3-(4,5-dimethylthiazol-2-yl)-2,5-diphenyltetrazolium bromide (MTT) assay. DMSO was used as a vehicle control. Data are means ±SD of at least three independent experiments. (**B**) BxPc-3 and AsPC-1 cells were treated with sulforaphane, **SF102** or **SF134** as described above, followed by plating of viable cells 24 h later at a low density (BxPc-3: 500 cells/well, AsPC-1: 1000 cells/well) in 6-well plates. After two weeks, colonies were fixed with 4% paraformaldehyde (PFA) and Coomassie-stained. Colonies containing more than 50 cells were quantified under a dissecting microscope and photographed. Representative images are shown on the right. The percentage of plating efficiency was calculated by setting the number of surviving colonies in the control group to 1. The diagrams on the left show the survival fraction (1st Gen). The surviving cells were harvested from non-fixed and non-Coomassie-stained duplicate plates of the first generation (1st Gen) of colony formation, and re-plated at the same density in 6-well plates. Two weeks later, the clonogenic survival of the second generation (2nd Gen) was evaluated as described above. The means ±SD are shown. ** *p* < 0.01, * *p* < 0.05.

**Figure 3 biomolecules-10-00769-f003:**
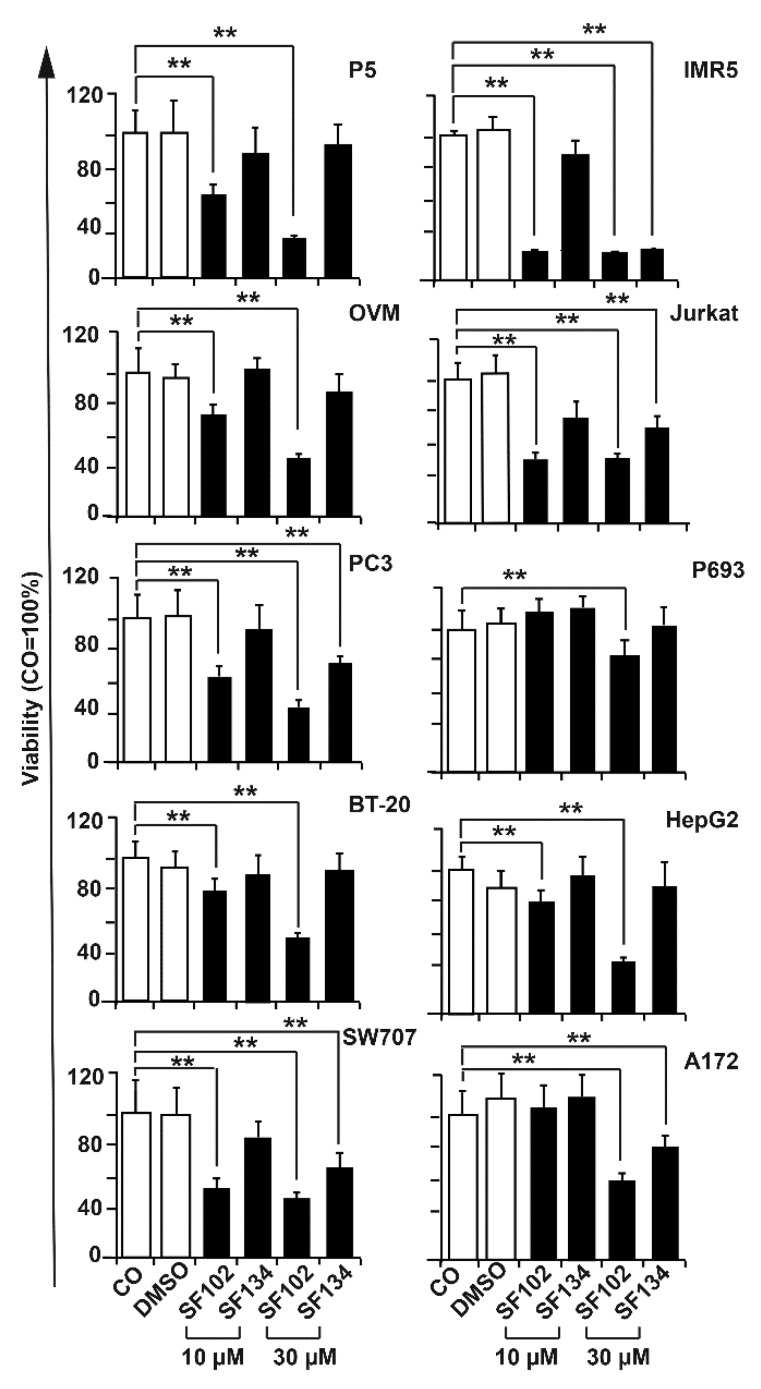
**SF102 and SF134 repress viability in various cancer entities.** Ovary (OVM), prostate (PC3), breast (BT-20), colorectal (SW707), glioblastoma (A172), hepatocellular (HepG2), neuroblastoma (IMR5), and T cell leukemia (Jurkat) cell lines were treated with **SF101** or **SF102** in concentrations of 10 µM and 30 µM. After 24 h, the viability was measured by MTT assay. DMSO served as a vehicle control. Data are means ±SD of at least three independent experiments. * *p* < 0.05, ** *p* < 0.01.

**Figure 4 biomolecules-10-00769-f004:**
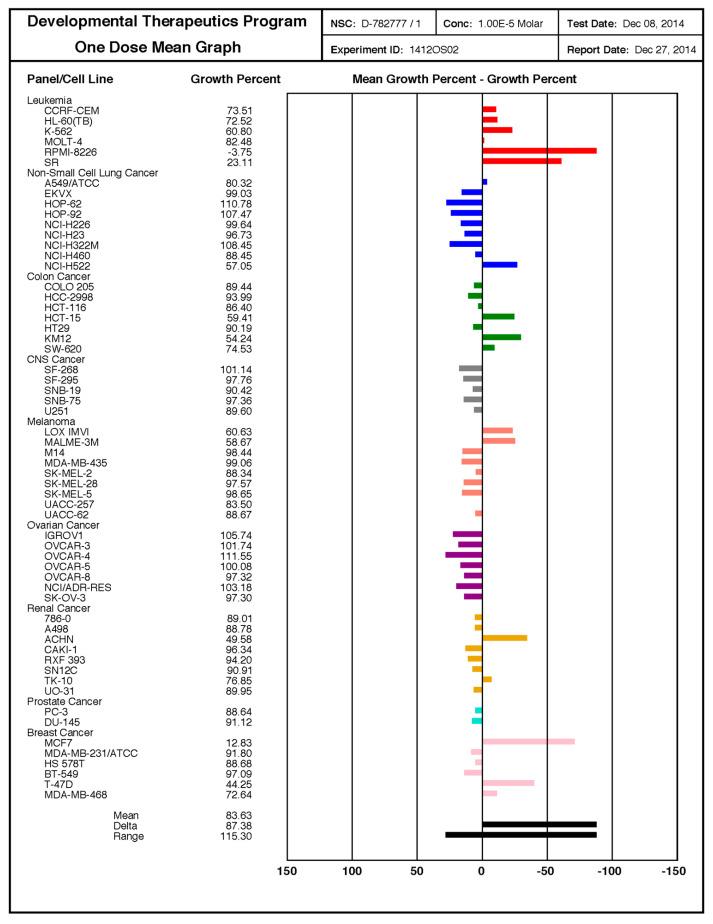
**SF102 is active in 60 cancer cell lines of the NCI-60 cell panel**. These studies were performed by the Services of the Discovery & Developmental Therapeutics Program at the NIH (http://dtp.cancer.gov). **SF102** was applied at a single dose (10 µM) to the 60 cancer cell lines representing leukemia, melanoma, non-small-cell lung carcinoma, and cancers of the brain, ovary, breast, colon, kidney, and prostate, as indicated. The growth inhibition (values between 0 and 100) and lethality (values less than 0) were measured. For example, a value of 100 means no growth inhibition. A value of 40 means 60% growth inhibition. A value of 0 means no net growth over the course of the experiment. A value of −40 means 40% lethality. A value of −100 means all cells are dead.

**Figure 5 biomolecules-10-00769-f005:**
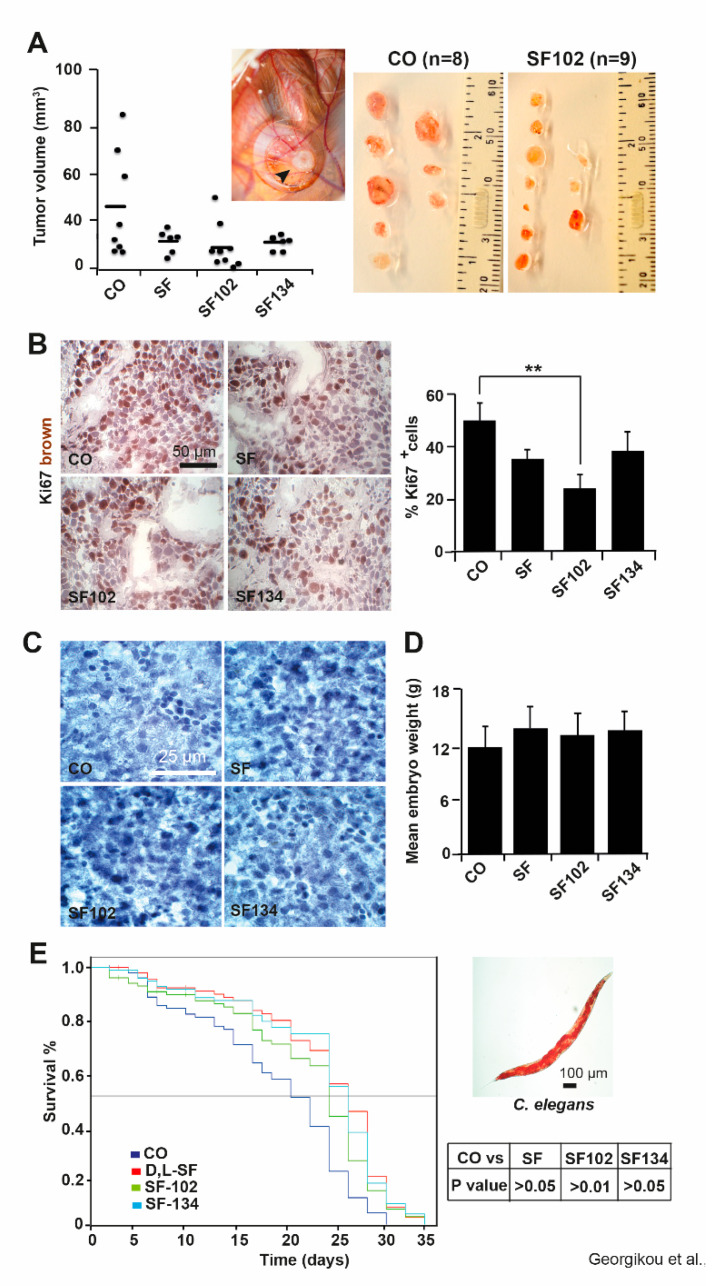
**SF102 and SF134 inhibit tumor xenograft growth without obvious side effects.** (**A**) BxPc-3 cells were seeded onto the CAM of fertilized chicken eggs on day 9 of embryonal development. Sulforaphane (SF, n = 6), **SF102** (n = 9), or **SF134** (n = 6), 10 µM each, were injected in 50 µl PBS into the CAM vessels on day 14 of development. The tumor xenografts were resected on day 18, and the tumor volumes were calculated as described in the Methods part. The control (CO, n = 8) group refers to xenografts that have received a PBS injection only. A representative image of a xenograft on the CAM (black arrow) on day 18 is shown. The volumes of individual tumors (black dots) and the mean tumor volumes of each experimental group (black lines) are shown. Statistical significance according to Bonferroni-Holm correction for multiple tests was not obtained and the single P values were: P = 0.646 for CO versus SF; P = 0.098 for CO versus **SF102**; P = 0.649 for CO versus **SF134**. (**B**) Frozen xenograft tissue sections were stained with the proliferation marker Ki67 (brown), and representative immunofluorescence pictures are shown on the left under 400× magnification. The scale bar indicates 50 µm. The intensity of the Ki67 immunofluorescence signal was quantified in 10 randomly chosen vision fields using ImageJ and the means ±SD are shown in the diagram on the right. ***p* < 0.01. (**C**) Hematoxylin staining of embryonal liver sections of embryos derived from fertilized chicken eggs, which were treated as described above. The scale bar indicates 25 µm. (**D**) The mean embryonal weight of the chick embryos was determined by weighting—the means of 15 embryos per group ±SD are shown. (**E**) Wild type *C. elegans* nematodes were treated with sulforaphane (SF), **SF102**, or **SF134** for 48 h, or were left untreated (CO) and Kaplan Meier analysis was performed. A representative microscopy image from a *C. elegans* worm is shown on the right. The scale bar indicates 100 µm.

**Figure 6 biomolecules-10-00769-f006:**
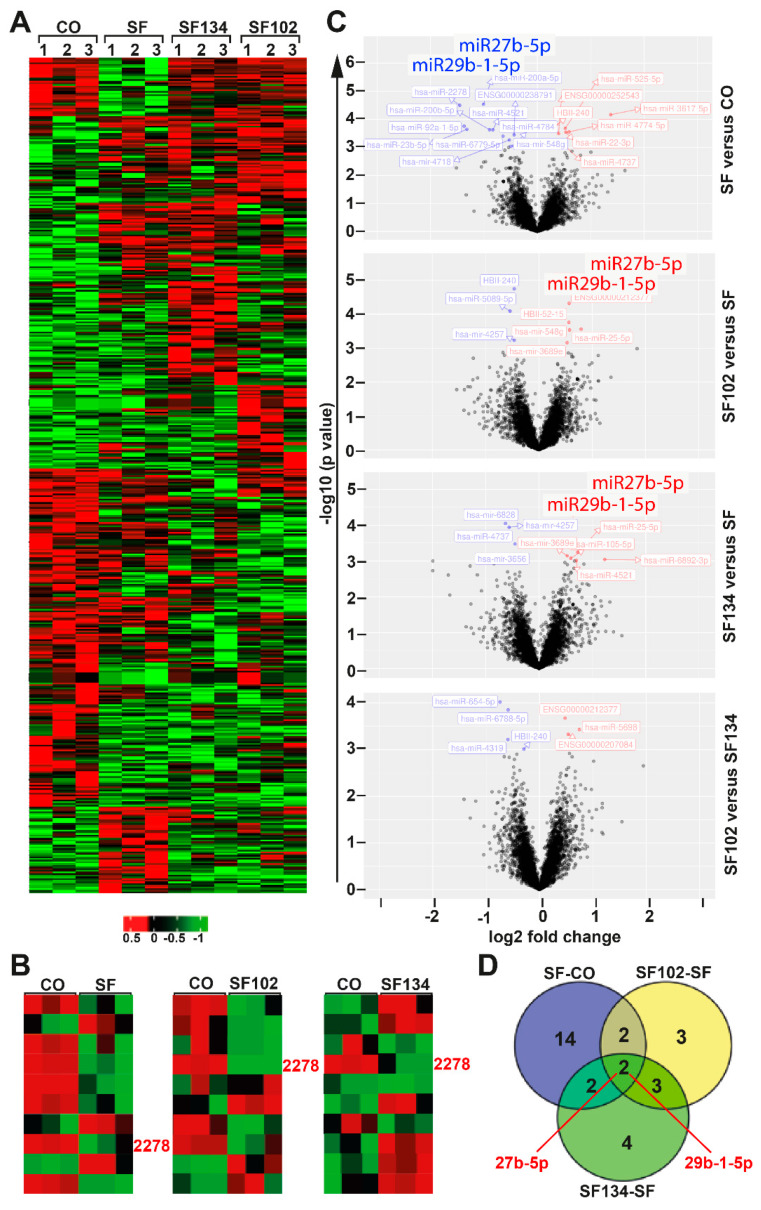
**miRNA expression profiling reveals differential expression induced by sulforaphane, SF102 or SF134.** (**A**) AsPC-1 cells were treated with 10 µM sulforaphane (SF), **SF102**, or **SF134,** or were left untreated (CO). After 24 h, RNA was isolated and analyzed by a GeneChip miRNA 4.0 Array in triplicates. The heatmap presents the top significantly regulated miRNAs. The red color marks high expression, and the green color marks low expression within a scale from 0.5 to –1, as indicated. (**B**) The 3 heatmaps present the top 10 significantly regulated miRNAs. miR2278 is highlighted in pink color. (**C**) Volcano plots show the miRNA distribution. On the y-axis, the –log10 p value is plotted, and on the x-axis, the fold change is plotted. (**D**) A Venn diagram shows the distribution of differentially expressed miRNAs among three groups: SF—CO, **SF102**—SF, and **SF134**—SF. The overlapping region identified two miRNAs, miR27b-5p and miR29b-1-5p, which are differentially expressed between sulforaphane, **SF102** and **SF134.**

**Figure 7 biomolecules-10-00769-f007:**
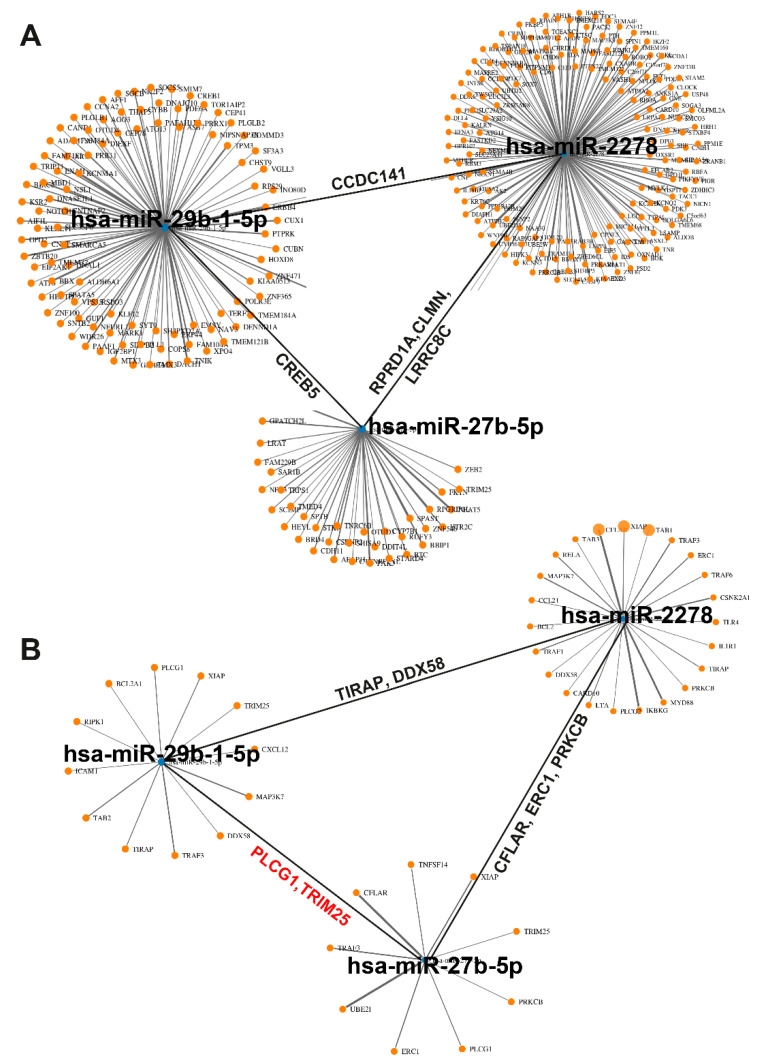
**Identification of NF-****κB-related miRNA-target genes.** (**A**) Using the Target Mining option of the mirWalk online platform, 309 target genes of miR2278, miR27b-5p, and miR29b-1-5p were identified and displayed as a network graph. Each orange dot is associated to a target gene. The black lines symbolize, e.g., the joint target gene CCDC141 of miR2278 and miR29b-1-5p, the joint target gene CREB5 of miR-29b-1-5p, and the joint target genes RPRD1A, CLMN, and LRRC8C of miR-2278 and miR27b-5p. (**B**) An miRNA-target gene analysis with focus to NF-κB signaling identified the most significantly NF-κB-related target genes of miR29b-1-5p, miR2278, and miR27b-5p. TIRAP and DDX58 are target genes of miR2278 and miR29b-1-5p, CFLAR, ERC1, and PRKCB are target genes of miR2278 and miR27b-5p, and PLCG1 and TRIM25 are target genes of miR29b-1-5p and miR27b-5p.

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
