# Peer review of "Novel Broccoli Sulforaphane-Based Analogues Inhibit the Progression of Pancreatic Cancer without Side Effects"

_biomolecules, 2020, doi:10.3390/biom10050769_

Round 1
Reviewer 1 Report
This manuscript from the research groups of Bolm and Herr report the synthesis and biological characterization of a series of novel analogs of sulforaphane. I have mixed impressions of the manuscript in its present form:
On one hand, I see a large value in the novel syntheses and compounds reported.
I am surprised by the finding that the compounds behave different from sulforaphane as the mode of action in both series is most likely related to the electrophilic isothiocyanate that reacts with cellular glutathione and thereby activates NF-kB signaling, i.e. the other part of the molecule is mainly a bystander. This is also partially verified, by the RNA expression analysis where the smaller differences seen could result from impurities, different exposures due to variation in compound solubility, etc, etc.
I am even skeptical to the conclusion drawn from the chicken embryo experiment (if I understand the model correctly). Regarding that experiment in Figure 5 – I question if the results presented here are really statistically significant. If I understand the assay correctly, only 3/8 tumors start to grow in the control? I.e. 5 are not growing even with just PBS treatment? Is this not surprising and suggest a failed seeding of the xenograft? Perhaps the effect seen with compounds are also just a result of 38% success rate in preparing the xenograft? Similarly, in panel B – in order to see the presented histology I assume 1 of the 3 positive xenograft were picked as positive control and not randomly from all 8 controls?
The conclusion section warns that the purity of compounds may affect the biological results. I understand this for sulforaphane isolated from natural sources, but here the compounds were prepared to what appears reasonable purity? It would be good to add a statement on the general purity of the compounds used in the assay, normally one would like to have 95% purity for biological assays; if lower, a statement to warn for this at an individual compound level would be preferred over the general disclaimer now presented at the last section of the paper.
I would also like to see a general disclaimer that the solubility of the compounds were not measured and may have impacted the activity of the compounds in the cellular studies; i.e. SF102 and SF134 might not be better than the other new compounds, but they might be more soluble under the assay conditions.
I recommend that the paper is revised based on the comments above, or that a more detailed explanation is presented.
“Minor” changes:
Graphics:
I need to inform that the pdf I downloaded had several technical problems with the drawings so I urge the authors and editorial staff to look extra carefully at the following illustrations to secure no minor errors that could have been caught by the referees are visible:
Figure 1 – impossible to see the synthesis scheme in my version but based on the figure legend I am able to follow the work. Needs to be polished and double checked to a usable version.
Figure 2A – the label on the x-axis is only seen for CO, SF, SF86 (to the left 10uM) and CO, SF, F85, and SF86 (to the right 30um). All compound names should be visible here for clarity.
Text:
Figure 4 – Legend – SF102 is effective in 60 cell lines …. I would not say it is very effective when it promotes growth of some cancer cell lines. A better word than “effective” would be “active”.
Competing interest and funding sections are duplicated.
Author Response
This manuscript from the research groups of Bolm and Herr report the synthesis and biological characterization of a series of novel analogs of sulforaphane. I have mixed impressions of the manuscript in its present form:
On one hand, I see a large value in the novel syntheses and compounds reported.
I am surprised by the finding that the compounds behave different from sulforaphane as the mode of action in both series is most likely related to the electrophilic isothiocyanate that reacts with cellular glutathione and thereby activates NF-kB signaling, i.e. the other part of the molecule is mainly a bystander. This is also partially verified, by the RNA expression analysis where the smaller differences seen could result from impurities, different exposures due to variation in compound solubility, etc, etc.
Comment 1:
I am even skeptical to the conclusion drawn from the chicken embryo experiment (if I understand the model correctly). Regarding that experiment in Figure 5 – I question if the results presented here are really statistically significant. If I understand the assay correctly, only 3/8 tumors start to grow in the control? I.e. 5 are not growing even with just PBS treatment? Is this not surprising and suggest a failed seeding of the xenograft? Perhaps the effect seen with compounds are also just a result of 38% success rate in preparing the xenograft? Similarly, in panel B – in order to see the presented histology I assume 1 of the 3 positive xenograft were picked as positive control and not randomly from all 8 controls?
Our response: We apologize for the lack of clarity and would like to explain that a high variability in xenograft growth is typical of an in vivo experiment - it is e.g. also observed with mouse xenografts. We would also like to point out that all of 8 xenografted tumors started to grow in the control, but 3 of them grew faster. Tumors, which did not start to grow on eggs were excluded from all groups before treatment. Also, we re-calculated the statistical significance and used this time the Bonferroni-Holm correction for multiple tests. This analysis resulted in no statistical significance of either group compared to the control group.
We also didn't want to give the impression that we picked the best data in Panel B. Rather, the figure on the left shows representative histogram pictures from each group, while the diagram on the right shows the mean values of Ki67-positive cells of ALL tumor xenografts of each group.
Change in the manuscript: Revision of Fig. 5 with additional data and text revision in the figure legend, discussion and results parts.
Comment 2:
The conclusion section warns that the purity of compounds may affect the biological results. I understand this for sulforaphane isolated from natural sources, but here the compounds were prepared to what appears reasonable purity? It would be good to add a statement on the general purity of the compounds used in the assay, normally one would like to have 95% purity for biological assays; if lower, a statement to warn for this at an individual compound level would be preferred over the general disclaimer now presented at the last section of the paper.
Our response: We fully agree to the reviewer's statement with respect to product purity, which we carefully analyzed after product preparation. All details had already been provided in the Supporting Information, where it was stated that "SF134 was delivered as one batch with ³99.5% purity, SF102 as three batches with 95.6%, 96.0% and 92.3% purity, respectively."
Change in the manuscript:
Text revision in the M&M section and explanation of the product purities.
Comment 3:
I would also like to see a general disclaimer that the solubility of the compounds were not measured and may have impacted the activity of the compounds in the cellular studies; i.e. SF102 and SF134 might not be better than the other new compounds, but they might be more soluble under the assay conditions.
Our response: We agree with the reviewer in asking for a comment on compound properties such as solubility and adapted the text respectively.
Change in the manuscript: Text revision in the discussion part.
Comment 4:
I recommend that the paper is revised based on the comments above, or that a more detailed explanation is presented.
Our response: We agree and have made the necessary changes.
Change in the manuscript: Figure and text revisions as explained in the points above.
“Minor” changes:
Graphics:
I need to inform that the pdf I downloaded had several technical problems with the drawings so I urge the authors and editorial staff to look extra carefully at the following illustrations to secure no minor errors that could have been caught by the referees are visible:
Figure 1 – impossible to see the synthesis scheme in my version but based on the figure legend I am able to follow the work. Needs to be polished and double checked to a usable version.
Figure 2A – the label on the x-axis is only seen for CO, SF, SF86 (to the left 10uM) and CO, SF, F85, and SF86 (to the right 30um). All compound names should be visible here for clarity.
Our response: We apologize for the bad quality of drawing. We believe it is possible that this was caused by the journal upload system, which reduces the size of the images significantly. Our original images have a high resolution.
Change in the manuscript: We provide now in addition a link to the original high quality drawings:
https://www.dropbox.com/sh/75mv3i4xejdgral/AADdDh6jidCcgMx08Dy1TCCna?dl=0
Text:
Figure 4 – Legend – SF102 is effective in 60 cell lines …. I would not say it is very effective when it promotes growth of some cancer cell lines. A better word than “effective” would be “active”.
Our response: We agree and revised the text as suggested.
Change in the manuscript: Text revision.
Competing interest and funding sections are duplicated.
Our response: We apologize and deleted one version.
Change in the manuscript: Text revision by deletion of the competing interest and funding declaration from page 14.
Reviewer 2 Report
The authors identified 2 sulforaphane analogues, which showed promising inhibitory properties in examined tumor entities. Findings are promising in developing new sulforaphane-related drugs.. The study was well designed and performed. Analyses methods were appropriate, manuscript was well written, figures and tables were well displayed, findings are solid and has clinical significance.
Author Response
The authors identified 2 sulforaphane analogues, which showed promising inhibitory properties in examined tumor entities. Findings are promising in developing new sulforaphane-related drugs.. The study was well designed and performed. Analyses methods were appropriate, manuscript was well written, figures and tables were well displayed, findings are solid and has clinical significance.
Our response: We thank the reviewer for this compliment.
Change in the manuscript: No changes necessary.
Round 2
Reviewer 1 Report
The changes and clarifications made by the authors are appropriate and the paper can now be accepted after the following sentence has been corrected:
Whereas sulforaphane SF102 and SF134 significantly reduced the mean tumor sizes compared to control xenografts, whereas sulforaphane and SF134 had an obvious tumor suppressing effect, although these differences were not significant (Fig. 5A).
I think something like the following is what is meant:
Although visual inspection showed an apparent tumor suppressing effect for sulforaphane and SF134, these differences were not significant (Fig. 5A).
Author Response
We thank the reviewer for this helpful suggestion and made the appropriate text change, see page 9 of the marked text "Georgikou_Text_marked" - the revised text is labeled in red.
